# Coupled Analysis of Hydrodynamic Responses of a Small Semi-Submersible Platform and a Large Floating Body

**Jianye Yang [1,*], Jun Yan [1,2], Yan Zhao [1,2], Jinlong Chen [1,3] and Heng Jin [4]**

1   Ningbo Institute of Dalian University of Technology, Ningbo 315016, China; yanjun@dlut.edu.cn (J.Y.); yzhao@dlut.edu.cn (Y.Z.); cjldut@163.com (J.C.)
2   State Key Laboratory of Structural Analysis for Industrial Equipment, Department of Engineering Mechanics, Dalian University of Technology, Dalian 116024, China
3   Jiangsu Hengtong Marine Cable Systems Co., Ltd., Suzhou 215537, China
4   School of Electromechanical and Energy Engineering, NingboTech University, Ningbo 315100, China; jinheng@nit.zju.edu.cn
*   Correspondence: yangjy_nbi@dlut.edu.cn

**Abstract:** This paper focuses on the hydrodynamic interaction between the large floating body and a small transfer platform in a jettyless floating transfer system. A high-order boundary-element method combined with a direct time-domain-solution method to calculate and analyze the hydrodynamic response of the small platform while berthed with the fixed large floating body and freely floating large body under wave action was applied and compared with the hydrodynamic response of a single small transfer platform. It was found that when the large floating body and the small transfer platform were both located on the leeward side, they had little mutual influence, and the hydrodynamic response agreed well with that of the single small transfer platform and the single large floating body. While the small platform was located on the leeward side of the large floating body, it significantly affected the hydrodynamic response of the platform, resulting in a significant reduction in horizontal motion and pitch motion, meaning that the shielding effect was significant. Within a certain range of wave periods, the heave-motion amplitudes can be significantly reduced, but beyond that range, they increase. Therefore, it is important to carefully consider the relative motions of LNG transport ships and small platforms when connecting them via aerial jumper pipes in jettyless floating transfer systems.

**Keywords:** hydrodynamic response; interaction; large floating body; small transfer platform; shielding effect

## 1. Introduction

The jettyless floating transfer system is a novel offshore liquefied natural gas (LNG)-transfer concept that entails ship-to-shore or ship-to-ship transfers. This innovative approach eliminates the need for quays or piers and instead utilizes floating flexible pipelines, a floating transfer platform, and a reel to connect the terminal to a floating carrier for offloading. Several companies, including Econnect Energy in Norway, Stena Power & LNG Solutions in Norway, and Houlder LNG Technology & Solutions in England, have already adopted this conceptual design scheme. Various systems, such as the jettyless transfer system [1], the autonomous transfer system [2], and the floating transfer terminal [3], have been developed to accomplish LNG transfer without a traditional jetty.

One noteworthy aspect of this system is the employment of a small floating-platform berth on a large floating ship. In order to guarantee the safe transmission of LNG, it is crucial to identify the hydrodynamic-response characteristics of the floating transfer platform when it docks with a large floating ship in the design of a jettyless floating transfer system. Obtaining the hydrodynamic-response characteristics of the floating transfer platform requires the analysis of the hydrodynamic coupling of multiple floating bodies.

The analysis of the hydrodynamic coupling of multiple floating bodies has been extensively explored by researchers, with existing studies primarily focusing on two aspects. The first pertains to large structures, such as very large floating bodies [4]. and "Pelamis" wave-energy devices [5], which entail multiple small floating bodies interconnected by constraints, with sufficient spacing between each of them. In this case, the small floating bodies are typically simple structures, like cylinders or boxes. The second aspect involves narrow gaps between slender structures, such as two vessels alongside each other.

When examining the hydrodynamic analysis of simple structures, like cylinders or boxes, Goo and Yoshida [6] developed a numerical approach using a three-dimensional source-distribution method and the interaction-linear-potential theory to forecast the wave-exciting forces and motions of two freely floating bodies in shallow water. In another study, Kagemoto and Yue [7] employed an exact algebraic method to investigate three-dimensional diffraction and radiation by several separate non-overlapping cylinders, using the linear-potential flow theory. Although this method can effectively solve the complete problem by predicting wave-exciting forces, hydrodynamic coefficients, and second-order-drift forces, it is exclusively based on the diffraction characteristics of a single cylinder. Subsequently, Kagemoto and Yue [8] utilized this method for very large floating structures with multiple legs by employing a matching concept that divided the entire structure into an interior core and a relatively small number of legs near the outer boundary. They solved the inverse-hydrodynamic-interaction problem and obtained optimal leg arrangements for minimal wave forces, displacements, and more. Siddorn and Eatock Taylor [9] also developed an exact algebraic method based on the boundary conditions satisfied by the sum of several Fourier–Bessel series for the combined radiation-and-diffraction problem. This method was applied to a square array of truncated cylinders, and it yielded the hydrodynamic coefficients of each cylinder, free surface elevations, and the excitation forces on each cylinder. In a 3D-time-domain approach by Zhu et al. [10], the influence of the gap on the wave forces in multiple floating structures was explored. The results showed that a sharp peak-force response appeared on each floating body for certain resonant wave numbers, and non-dimensional in-line wave forces were present on both the leading and the shielded floating bodies. The study revealed that the in-line wave force on the leading floating body was greater than that on the shielded floating body, and the shielding effect of the leading floating body became important. Huang et al. [11] analyzed the dynamic response of a floating bridge under the two conditions of "with a floating platform" and "without a floating platform." The results showed that the floating platform served as a wave shield, reducing the motions in the heave, surge, and pitch of the floating bridge. Although the shielding effects of the platform decreased the longitudinal force of the connectors to some extent, the vertical force on these connectors was minimally impacted.

In Li's study [12], the hydrodynamic interaction between two vessels was explored, focusing on the resonant characteristics in parallel and nonparallel configurations for a real hull-shaped FPSO and a ship, with different wave headings analyzed using Cummins' approach in the time domain. Numerical and experimental evaluations of the shielding effect were conducted by switching vessels on the lee and weather sides. The results indicated that distinct degrees of freedom tended to react to resonant modes, with the higher-resonance mode shifting to a lower frequency in nonparallel configurations. The shielding effect only suppressed the motion caused by the gap resonance, while the natural-frequency-resonance-like roll remained unaffected. The influence of the weather-side exchange between the ship and the FPSO regarding the hydrodynamic interaction for the FPSO was negligible, due to significant differences in size and volume between the two ships. Sun et.al. [13] studied the wave-induced responses of constrained multiple bodies by applying the linear-diffraction theory and imposing constraints on the body connections using the Lagrange multiplier technique. Two cases involving two rectangular boxes connected by a hinge or rigid rod were investigated, and a tanker was considered alongside an FLNG barge. The study found that the Lagrange multiplier technique was convenient for analyzing multiple rigid bodies connected by rigid or flexible connections, and that the

behaviors of the vessels with rigid or hinged horizontal connections were generally similar. Feng and Bai [14] applied a nonlinear decomposition model with the potential-flow theory to investigate the hydrodynamic performances of two freely floating or interconnected barges. The results indicated that in the case of the hinge connection, potentially large constraint moments were relieved. In the symmetric configuration in a head sea, the yaw drift was reduced in the case of a middle-hinge connection compared to two freely floating barges. In a beam sea, the hinge connections aided in diminishing the discrepancy in motion between the windward and lee-side barges, suppressing the motion of the windward barge while enlarging that of the lee-side barge. Li [15] investigated the hydrodynamic coupling of a semi-submersible floating wind turbine and an offshore support vessel during walk-to-work operations, using 3D diffraction and radiation computation in the frequency domain. Ignoring the hydrodynamic interaction led to over-prediction of motion of the support vessel in surge and sway due to the overestimation of the drift force. The sway and roll of the wind turbine decreased significantly, up to 40%, due to a shielding effect, which cut down both the linear and the nonlinear wave forces in the entire frequency range. The sway and heave of the support vessel displayed gap-resonance behavior, where different modes of peak or trough were observed, whose occurrence was opposite to sway and heave. A higher-order boundary-element method (HOBEM) combined with a generalized mode approach was applied to the analysis of the motion and drift force of multiple side-by-side-moored vessels with small gaps by Hong et al. [16]. The numerical results were used to predict the total wave-drift force, even in the Helmholtz resonance frequency, and the wave-drift force was not significantly influenced by the roll-resonance phenomena captured in the measured relative wave at mid-ship of the LNG FPSO, with LNGCs moored alongside each other. The strength of the interaction decreased as the heading angle changed from the beam sea to the head sea. Kim [17] conducted a comparative study using numerical calculations of and experiments on the effects of hydrodynamic interactions in multi-body systems, using the LNG FPSO and a shuttle tanker. Both side-by-side and tandem mooring were considered. In the tandem mooring, the shielding effect was noticeable in the drift force, while the distance had no significant effect on the longitudinal force. In the side-by-side mooring, the lee-side vessel's shielding effect on the drift force and motion RAO was significant, with the lee-side ship acting as a block to disturb the wave-flow pattern laterally. The closer the distance between the vessels, the greater the reciprocally amplified magnitude of the lateral drift. Kuriakose [18] compared the hydrodynamic performance of a single body and multiple bodies near each other. They found significant hydrodynamic interactions, resulting in forces and responses which were up to double those of the single-body case. Additionally, there was a shielding effect on the responses on the leeward-side body. Wolgamot et al. [19] considered waves radiated from circular arrays of truncated cylinders oscillating independently in still water. Body motions that excited the same free-surface motion local to the array as the sloshing near-trapped mode were associated with enhanced radiation from the array. The authors suggested that rigid body-pumping modes might be relevant to structures like tension-leg platforms or semi-submersible platforms, with arrays of cylinders forced to move together.

A jettyless floating transfer system involves two floating bodies: an LNG carrier, and a small floating transfer platform for transmission operations. These two bodies have significant size differences. Fæhn [20] presented a universal buoyancy system for LNG offloading for small-scale ships and applied SIMA software to investigate the behaviors of the platform alone, the platform connected to the ship side, and the complete system with a pipeline, ship, and platform. The results indicated that the complete system had the largest recordings for all the wave headings, and the pipeline added a net excitation to the system. However, the analysis of the hydrodynamic coupling of the platform and the ship was incomplete, and the impact of the ship's shielding effect on the platform's motion and forces was unknown.

This paper reports the use of a direct time-domain high-order boundary-element method [21,22] using the linear-potential-flow theory to study the hydrodynamic coupling

problem of a small semi-submersible platform and a large floating box. One integral equation and two equations of motion are solved without considering the connection between the two floating bodies. The aim is to investigate the shielding effect of the large floating box on the motion responses of the small platform, both when the large floating box is fixed and when it is freely floating under wave action. In particular, the relative motion amplitudes between the platform and the large box are shown, providing important design references for a jettyless floating transfer system due to the presence of airborne spanning pipes between the platform and the hull.

## 2. Theory and Numerical Model

This study adopts a higher-order boundary-element method based on the diffraction-and-radiation theory. The body's boundary condition is satisfied on the wetted body surface at static equilibrium, while the linear free-surface boundary condition is applied on the still-water surface. The aim of this research is to investigate the impact of diffraction and radiation of a large floating body on the motion responses of a small semi-submersible platform.

### 2.1. The Initial Boundary-Value Problem of Velocity Potential

To solve the 3D diffraction-and-radiation problem, a right-handed Cartesian coordinate system, $Oxyz$, is introduced, as shown in Figure 1. The $x$–$y$ plane coincides with the mean free surface, and the $z$-axis is positive upward and passes through the center of gravity of the small floating body in static equilibrium. Additionally, the right-handed Cartesian coordinate system, $O_1x_1y_1z_1$, is fixed on the large floating body. The $x_1$–$y_1$ plane coincides with the mean free surface, and the $z_1$ axis is positive upward and passes through the center of gravity of the large floating body in static equilibrium. The fluid is assumed to be inviscid, homogeneous, impressible, and irrotational. In Figure 1, $S_F$ represents the free-water face, $S_{BL}$ is the surface of the large floating body, $S_{BS}$ represents the body surface of the small floating body, and $S_D$ is the seabed.

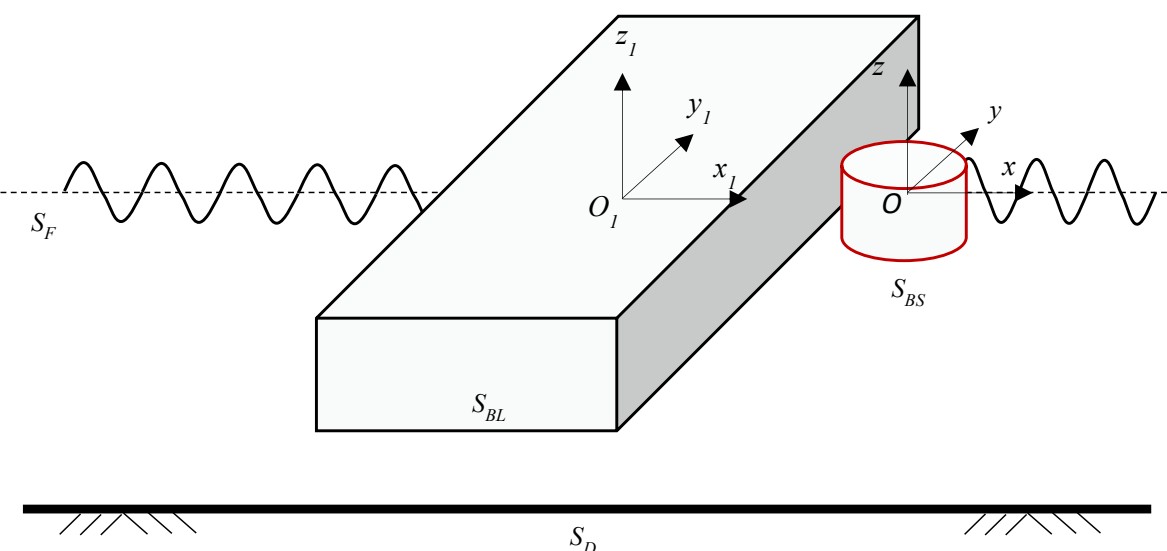

**Figure 1.** Sketch of the definition.

The velocity potential $\Phi(x, y, z, t)$ and wave-surface function $\eta(x, y, z, t)$ can be written as

$$\Phi = \Phi_I + \Phi_S \tag{1}$$

$$\eta = \eta_I + \eta_S \tag{2}$$

where the subscripts $I$ and $S$ represent the incident wave and scattering wave, respectively.

The scattering-wave problem can therefore be formulated in terms of a velocity potential $\Phi_S(x, y, z, t)$ that satisfies the Laplace equation in the fluid domain,

$$\nabla^2 \Phi_S = 0 \tag{3}$$

The scattering-velocity potential is also subject to various boundary conditions on all surfaces of the fluid domain. The linearized free-surface boundary conditions are

$$\frac{\partial \eta_S}{\partial t} = \frac{\partial \Phi_S}{\partial z}, \quad \text{on} \quad S_F \tag{4}$$

$$\frac{\partial \Phi_S}{\partial t} = -g\eta_S, \quad \text{on} \quad S_F \tag{5}$$

where $\eta_S$ is the free surface elevation, $g$ is the acceleration due to gravity, and $S_F$ is the mean free-water surface.

The boundary condition on the seabed satisfies the impermeable condition

$$\frac{\partial \Phi_S}{\partial n} = 0, \quad \text{on} \quad S_D \tag{6}$$

For infinite water-depth problems, Equation (5) is written as

$$\nabla \Phi_S \to 0, \quad z \to -\infty \tag{7}$$

The initial conditions for the present problem in the fluid domain can be written as

$$\Phi_S = \Phi_{St} = 0, \quad \text{at} \quad t = 0 \tag{8}$$

The body-boundary condition is

$$\frac{\partial \Phi_S}{\partial n} = \mathbf{U}_S \cdot \mathbf{n}_S + \boldsymbol{\omega}_S \cdot (\mathbf{r}_S \times \mathbf{n}_S), \quad \text{on} \quad S_{BS} \tag{9}$$

where $\mathbf{U}_S$ and $\boldsymbol{\omega}_S$ are the body translation and rotation velocity of the small floating body, respectively. The $\mathbf{r}_S$ is the position vector from the rotation center to a point on the small floating body's surface, $\mathbf{n}_S$ the unit normal vector, perpendicular to the surface of the small floating body pointing towards the fluid domain.

(1)　Only considering the diffraction of the large floating body

$$\frac{\partial \Phi_S}{\partial n} = 0, \quad \text{on} \quad S_{BL} \tag{10}$$

(2)　Considering the diffraction and radiation of the large floating body

$$\frac{\partial \Phi_S}{\partial n} = \mathbf{U}_L \cdot \mathbf{n}_L + \boldsymbol{\omega}_L \cdot (\mathbf{r}_L \times \mathbf{n}_L), \quad \text{on} \quad S_{BL} \tag{11}$$

where $\mathbf{U}_L$ and $\boldsymbol{\omega}_L$ are the body translation and rotation velocity of the large floating body, respectively. The $\mathbf{r}_L$ is the position vector from the rotation center to a point on the large floating body's surface, $\mathbf{n}_L$ the unit normal vector, perpendicular to the surface of the large floating body pointing towards the fluid domain.

### 2.2. Integral Equation

To solve the initial boundary-value problem, a time-stepping method is employed. In each time step, the radiation-velocity potential that satisfies the Laplace equation is calculated using an integral-equation method. By applying Green's second identity in the

fluid domain, the boundary-value problem described above is converted into the following boundary integral equation in a conventional manner (Teng et al. [23]).

$$
\begin{aligned}
\alpha(\mathbf{x}_0)\Phi_S(\mathbf{x}_0,t) &= \iint\limits_{S_B+S_F} \left[ G(\mathbf{x},\mathbf{x}_0)\frac{\partial \Phi_S(\mathbf{x},t)}{\partial n} - \Phi_S(\mathbf{x},t)\frac{\partial G(\mathbf{x},\mathbf{x}_0)}{\partial n} \right] dS \\
&= \iint\limits_{S_{BL}+S_{BS}+S_F} \left[ G(\mathbf{x},\mathbf{x}_0)\frac{\partial \Phi_S(\mathbf{x},t)}{\partial n} - \Phi_S(\mathbf{x},t)\frac{\partial G(\mathbf{x},\mathbf{x}_0)}{\partial n} \right] dS
\end{aligned}
\tag{12}
$$

where $\mathbf{x}_0$ $(x_0, y_0, z_0)$ and $\mathbf{x}$ $(x, y, z)$ are the source and field points, respectively. The $\alpha(\mathbf{x}_0)$ is the free term, named the 'solid angle,' at the source point $\mathbf{x}_0$, and $G(\mathbf{x},\mathbf{x}_0)$ is a simple Green function of the Rankine source, and its image about the seabed, as

$$
G(\mathbf{x},\mathbf{x}_0) = -\frac{1}{4\pi}\left( \frac{1}{R_1} + \frac{1}{R_2} \right)
\tag{13}
$$

in which

$$
\begin{cases}
R_1 = \sqrt{\left( (x-x_0)^2 + (y-y_0)^2 + (z-z_0)^2 \right)} \\
R_2 = \sqrt{\left( (x-x_0)^2 + (y-y_0)^2 + (z+z_0+2d)^2 \right)}
\end{cases}
\tag{14}
$$

and $d$ is the water depth.

By incorporating an artificial damping zone at the outer-circular-ring cirque on the free-water surface, the integration on the initial infinite free surface can be confined to a smaller domain around the body. The setting method of the artificial damping zone can be found in the study by Ferrant [24] and the parameter settings can be found in the study by Bai and Teng [25]. After discretizing the integral Equation (8) with higher-order elements based on quadratic shape functions, a set of linear equations can be set up to determine the scattering potential on the body surface and the normal derivative of the velocity potential on the free surface. The discrete process and specific solution method of integral equation can be found in the study by Geng [26].

To update the free-surface elevation and velocity potential on the free surface, the free-surface boundary conditions are applied in a time-matching manner using the standard fourth-order Runge–Kutta scheme. The updated free-surface elevations and the velocity potentials are subsequently utilized to solve the mixed-boundary-value problem in the following time step.

*2.3. Hydrodynamic Forces*

Without considering the interaction of the connection between the small floating body and the large floating body, the interaction between their hydrodynamic responses is examined. The hydrodynamic force $\mathbf{F} = \{F_{1j}, F_{2j}, F_{3j}\}$ and moment $\mathbf{M} = \{M_{1j}, M_{2j}, M_{3j}\}$ acting on a body can be obtained by integrating the fluid pressure over the wetted body's surface

$$
F_{ij} = \iint\limits_{S_{Bj}} p n_{ij} dS = -\rho \iint\limits_{S_{Bj}} \left( \frac{\partial \Phi}{\partial t} + gz \right) n_{ij} dS, \quad (i=1,2,\cdots,6, \quad j=1,2)
\tag{15}
$$

where $p$ is the pressure of the body surface, $\rho$ is the density of the fluid, and $n_i$ is the $i$th component of the normal direction of the small or large floating body's surface, with the definition of $(n_{1j}, n_{2j}, n_{3j}) = \mathbf{n}_j$ and $(n_{4j}, n_{5j}, n_{6j}) = \mathbf{r}_j \times \mathbf{n}_j$. The $j = 1$ represents the small floating body and $j = 2$ represents the large floating body.

*2.4. Motion Equation*

The motion equations of two floating bodies without connections can be written as

$$
\sum_{i=1}^{6} \left[ \mathbf{M}_{mj,j}\ddot{\xi}_{i,j}(t) + \mathbf{B}_{mi,j}\dot{\xi}_{i,j}(t) + \left( \mathbf{C}_{mi,j} + \mathbf{K}_{mi,j} \right)\xi_{i,j}(t) \right] = \mathbf{F}_{m,j} \quad (m=1,2,\cdots,6, \quad j=1,2)
\tag{16}
$$

where **M** is a 6 × 6 matrix whose diagonal values are the body mass and rotational inertia. The **B** is a 6 × 6 viscous damping matrix, **C** is a 6 × 6 matrix, which represents the still-water-restoring-force matrix, and **K** is a 6 × 6 matrix, which represents the stiffness matrix. As $m$ = 1, 2, 3, $\mathbf{F}_{m,j}$ is the wave force acting on the small floating body ($j$ = 1) or the large floating body ($j$ = 2) along the $x$, $y$, and $z$ axes, respectively, and as $m$ = 4, 5, 6, $\mathbf{F}_{m,j}$ is the wave moment acting on the small floating body ($j$ = 1) or the large floating body ($j$ = 2) around the $x$, $y$, and $z$ axes, respectively. The $\mathbf{F}_{m,j}$ can be obtained by Equation (15). The $\xi$ is the displacement of the floating body, $\dot{\xi}$ is the velocity, and $\ddot{\xi}$ is the acceleration.

## 3. Model Validation

Goo and Yoshida [6] conducted a study on the multi-body hydrodynamics of a cylinder and a box. In this paper, we present a coupled analysis of the hydrodynamic responses of a cylinder and a box using the numerical calculation method described above. We then compare our results with Goo and Yoshida's results to demonstrate the feasibility of the current numerical calculation method.

### 3.1. Analysis Model of the Cylinder and Box

The detailed parameters of the cylinder and the box are shown in Table 1. The diameter of the cylinder is 95.8 m, the length of the box is 109.7 m, and the width is 101.4 m. The distance between the centroid of the cylinder and the centroid of the box is 102.75 m. The wave propagates from the box to the cylinder, with one side of the box formed by its width and height on the wave-facing side, and the other side as the opposite side, while the cylinder is positioned on the opposite side of the box.

**Table 1.** Parameters of the cylinder and box.

|  | **Cylinder** | **Box** |
|---|---|---|
| Length/m | - | 109.70 |
| Width/m | - | 101.40 |
| Diameter/m | 95.80 | - |
| Draught/m | 30.00 | 30.00 |
| Displacement/m$^3$ | 216,200 | 333,700 |
| Center of gravity above base/m | 29.90 | 29.80 |
| Longitudinal radius of gyration/m | 31.20 | 30.30 |
| Transverse radius of gyration/m | 31.20 | 30.40 |

### 3.2. Results of Model Validation

Eight-node quadrilateral elements were used to mesh the wet surface of the cylinder and the box, and the free-water surface. The number of mesh elements in the box was 220; for the cylinder, this number was 192, and for the free-water surface, it was 4078.

Figures 2 and 3 depict the surge and heave amplitudes of the cylinder and box, respectively, at wave frequencies of 0.3 rad/s, 0.35 rad/s, 0.45 rad/s, 0.5 rad/s, 0.55 rad/s, and 0.65 rad/s. These results agree well with Goo and Yoshida's results, irrespective of the surge or heave-motion amplitude. This indicates that the presented method can be used for the coupled analysis of the hydrodynamic responses of two floating bodies.

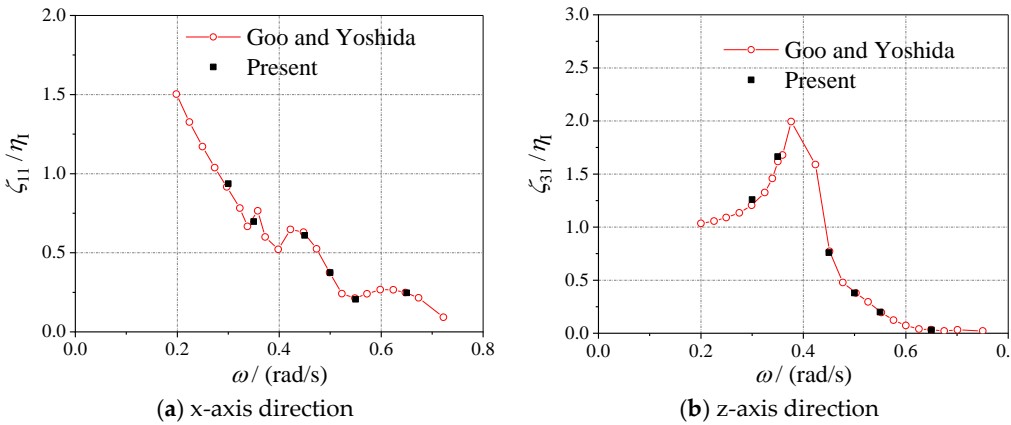

**Figure 2.** Motion responses of the box.

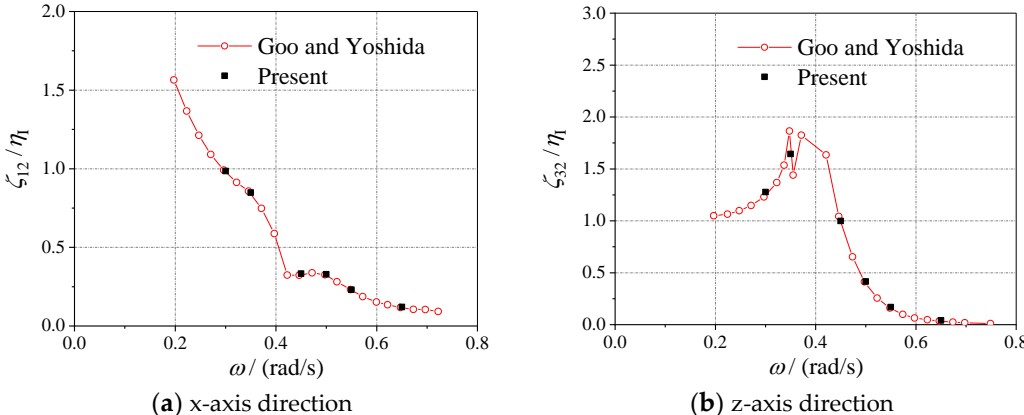

**Figure 3.** Motion responses of the cylinder.

## 4. Coupled Analysis of a Small Platform and a Large Floating Body

The jettyless floating transfer system involves two floating bodies, an LNG carrier, and a small floating transfer platform, for transmission operations. These bodies are equivalent to a large floating box and a small semi-submersible platform, respectively. In this section, the aim is to perform a hydrodynamic coupling analysis of the two bodies to investigate the issues in a jettyless floating transfer system, such as the shielding effect of the large floating box and hydrodynamic coupling effect.

### 4.1. Parameters of the Small Platform and the Large Floating Box

The parameters of the small semi-submersible platform and the large floating box are shown in Table 2. The ratio of the length of the semi-submersible platform to the large floating box was 1:6, the ratio of the width was 0.75:1, and the ratio of the draught was 0.77:1. The distance between the semi-submersible platform and the large floating box was 2.0 m. The layout plans of the two floating bodies are shown in Figure 4 and a plane dimensional drawing of the semi-submersible platform is shown in Figure 5.

The linear regular incident wave is adopted

$$\eta_I = A_I \cos\left[k_I\left(x\cos\alpha + y\sin\alpha - \frac{2\pi}{T}t + \varphi_0\right)\right] \tag{17}$$

where $A_I$ is the amplitude of the incident wave, $k_I$ is the wave number of the incident wave, $T$ is the period of the incident wave, $\alpha$ is the wave-incidence angle, $(x, y)$ are the coordinates of any point on the water surface, and $\varphi_0$ is the initial phase, $\varphi_0 = 0$.

**Table 2.** Parameters of the semi-submersible platform and the floating box.

|  | Semi-Submersible | Large Floating Box |
|---|---|---|
| Length/m | 15.0 | 90.0 |
| Breadth/m | 11.3 | 15.0 |
| Diameter of the column/m | 1.20 | - |
| Draught/m | 3.30 | 4.30 |
| Height of the submerged pontoon/m | 0.63 | - |
| Displacement/m$^3$ | 39.82 | 333,700 |
| Center of gravity above base/m | 2.62 | 4.20 |
| The longitudinal radius of gyration/m | 6.60 | 26.1 |
| Transverse radius of gyration/m | 4.60 | 4.8 |
| The vertical radius of gyration/m | 5.00 | 26.3 |

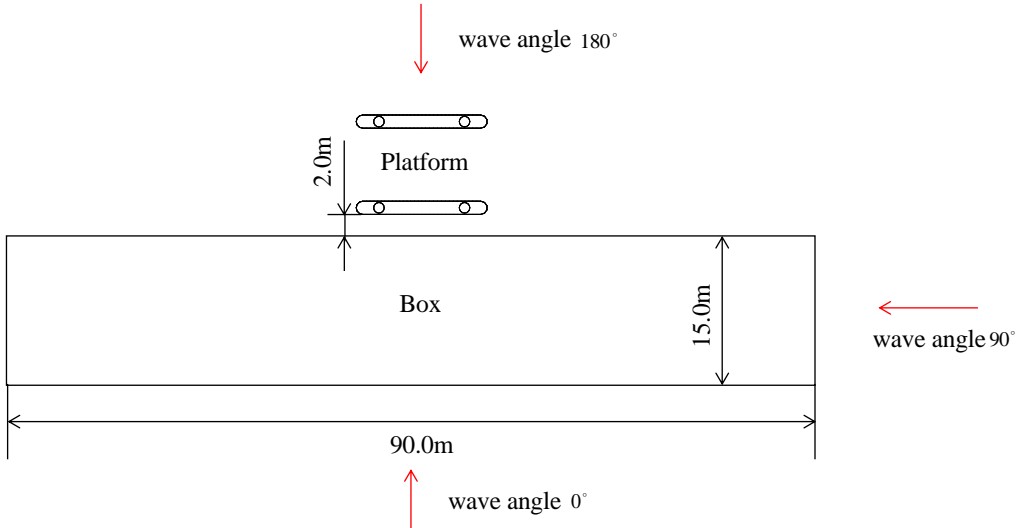

**Figure 4.** Layout plan of the large box and the semi-submersible platform.

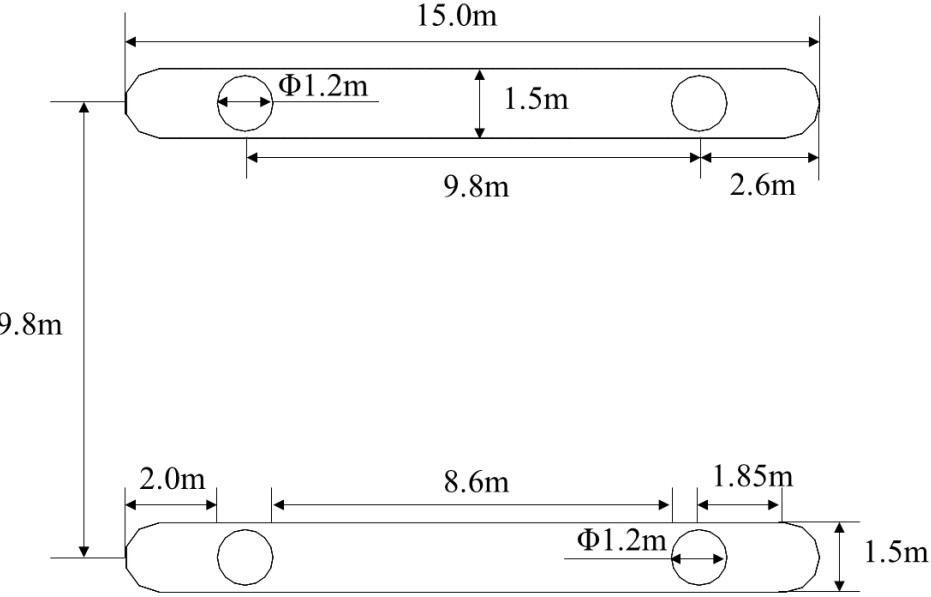

**Figure 5.** Plane dimensional drawing of the platform.

The diagonal element of the damping matrix of the small semi-submersible platform can be written as

$$b_{ii} = 2\zeta_i \omega_{0i} M_{ii} \quad (i = 1, 2, 3, \ldots, 6) \tag{18}$$

where $\omega_{0i}$ is the resonance frequency in the *i*-direction, $M_{ii}$ is the mass or moment-of-inertia mass in the i-direction, and $\zeta_i$ is the damping ratio.

The values of $\zeta_i$ for the large floating box are $\zeta_1 = \zeta_2 = 0.3$, $\zeta_3 = 0.125$, $\zeta_4 = 0.04$, $\zeta_5 = 0.08$ and $\zeta_6 = 0.1$, respectively.

The damping matrix of the small semi-submersible platform $\mathbf{B}_1$ is

$$\mathbf{B}_1 = \begin{bmatrix} 1.0 \times 10^4 & 0 & 0 & 0 & 0 & 0 \\ 0 & 1.0 \times 10^4 & 0 & 0 & 0 & 0 \\ 0 & 0 & 8.0 \times 10^3 & 0 & 0 & 0 \\ 0 & 0 & 0 & 8.0 \times 10^4 & 0 & 0 \\ 0 & 0 & 0 & 0 & 2.0 \times 10^5 & 0 \\ 0 & 0 & 0 & 0 & 0 & 2.0 \times 10^5 \end{bmatrix} \tag{19}$$

The stiffness matrix of the small semi-submersible platform $\mathbf{K}_1$ is

$$\mathbf{K}_1 = \begin{bmatrix} 1.0 \times 10^4 & 0 & 0 & 0 & 0 & 0 \\ 0 & 1.0 \times 10^4 & 0 & 0 & 0 & 0 \\ 0 & 0 & 0 & 0 & 0 & 0 \\ 0 & 0 & 0 & 1.0 \times 10^5 & 0 & 0 \\ 0 & 0 & 0 & 0 & 1.0 \times 10^5 & 0 \\ 0 & 0 & 0 & 0 & 0 & 1.0 \times 10^5 \end{bmatrix} \tag{20}$$

The still-water-restoring-force matrix of the small semi-submersible platform $\mathbf{C}_1$ is

$$\mathbf{C}_1 = \begin{bmatrix} 0 & 0 & 0 & 0 & 0 & 0 \\ 0 & 0 & 0 & 0 & 0 & 0 \\ 0 & 0 & 4.43 \times 10^4 & 0 & 0 & 0 \\ 0 & 0 & 0 & 3.64 \times 10^5 & 0 & 0 \\ 0 & 0 & 0 & 0 & 3.64 \times 10^5 & 0 \\ 0 & 0 & 0 & 0 & 0 & 0 \end{bmatrix} \tag{21}$$

*4.2. Meshing*

To mesh the wet surface of the small semi-submersible platform and the large floating body (LFB), as well as the free-water surface, eight-node quadrilateral elements were used. The mesh-generation process for the floating bodies and the free-water surface is shown in Figures 6 and 7, respectively. The total number of the mesh elements in the semi-submersible platform was 2266, and for the large floating box, it was 996. The number of mesh elements for the free-water surface was 5856.

As a result of the substantial size discrepancy between the two floating bodies, the meshing surrounding the small semi-submersible platform was relatively fine, while the meshing surrounding the large floating box was relatively rough; however, all the meshes provided the required calculation accuracy.

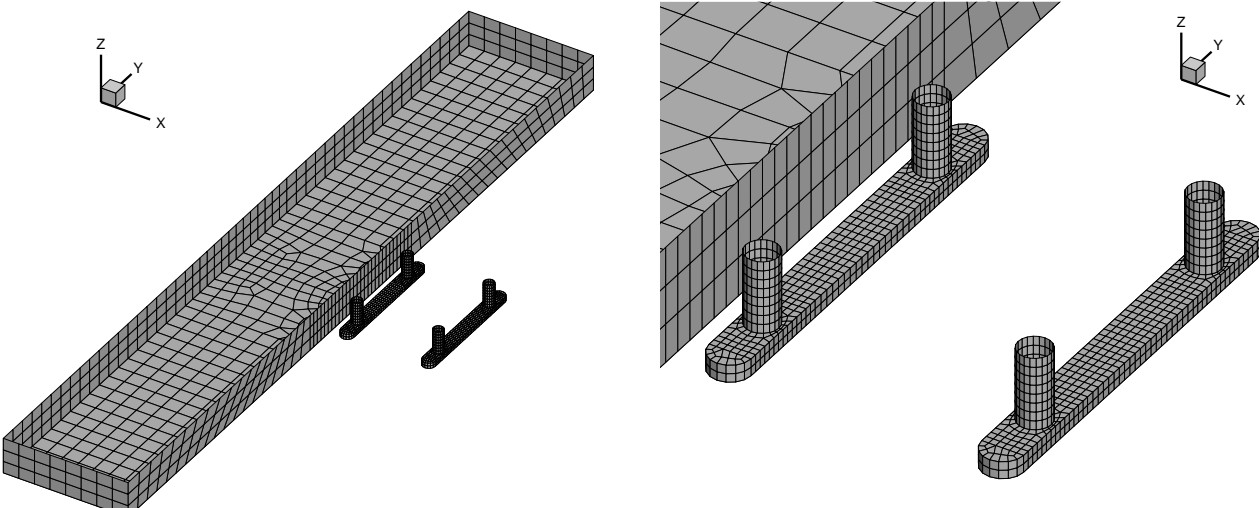

(**a**) Whole wet-surface meshes.　　　　　　(**b**) Local wet-surface meshes.

**Figure 6.** Meshes of the small semi-submersible platform and large floating box.

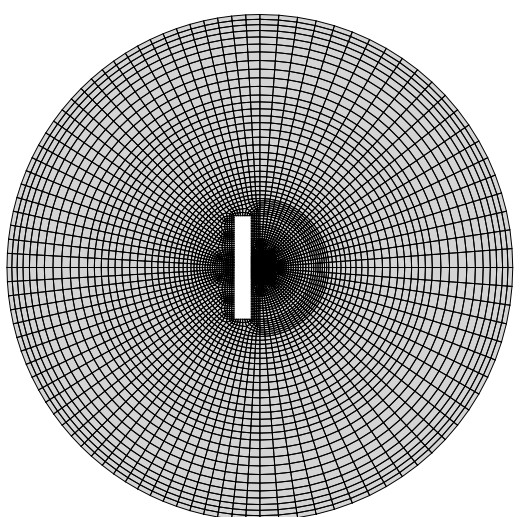

(**a**) Whole-water-surface meshes.　　　　(**b**) Local-water-surface meshes.

**Figure 7.** Meshes of the water surface.

*4.3. Consideration of the Motion Response of the Semi-Submersible Platform Only*

The large floating body was fixed, considering only the impact of its diffraction wave on the motion response of the small platform. The motion responses of the small platform for when the wave-incidence angles were 0°, 45°, 90°, and 135° (the definition of the incidence angle can be seen in Figure 4), and the wave period ranged from 3.5 s to 11.0 s, considering the wave conditions of China's seas [27], are shown in Figures 8–10.

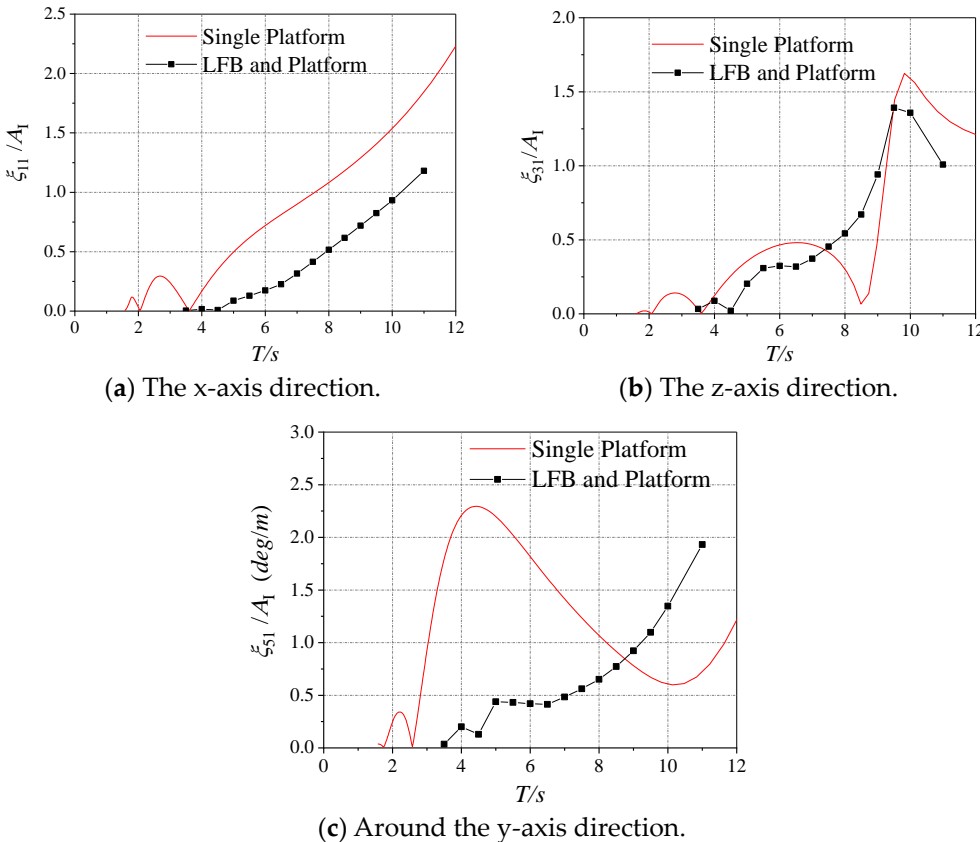

(**a**) The x-axis direction.

(**b**) The z-axis direction.

(**c**) Around the y-axis direction.

**Figure 8.** Motion responses of the small platform at wave angle $\alpha = 0°$.

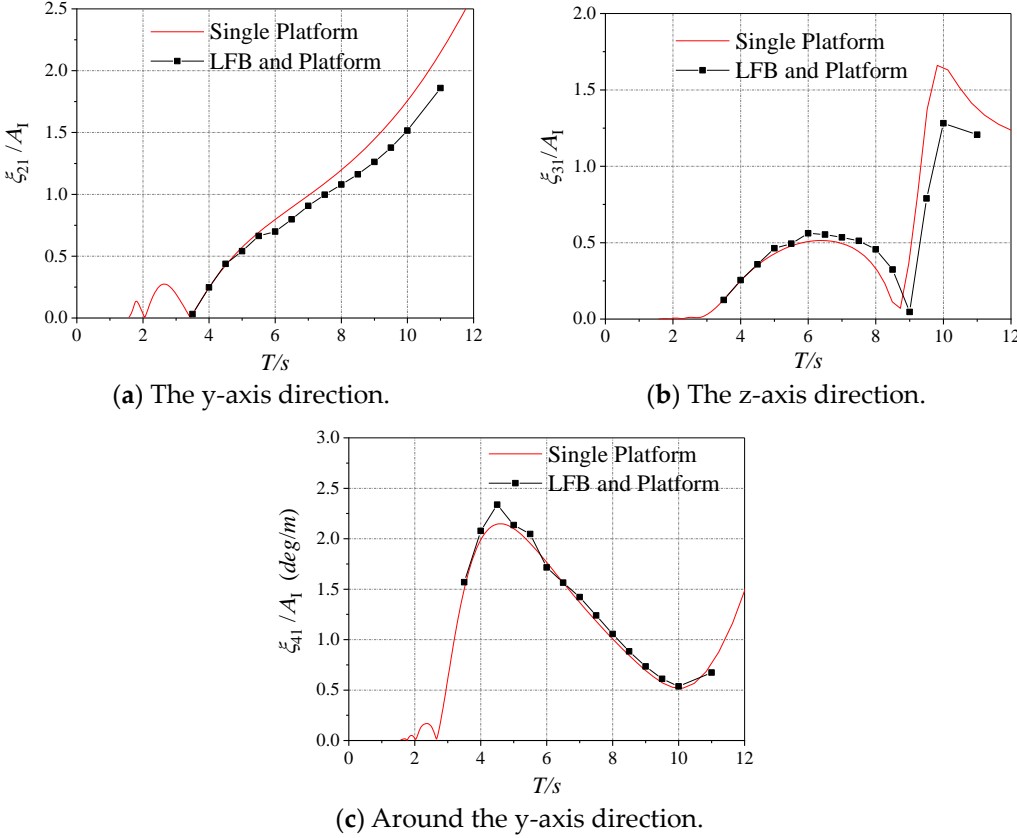

(**a**) The y-axis direction.

(**b**) The z-axis direction.

(**c**) Around the y-axis direction.

**Figure 9.** Motion responses of the small platform at wave angle $\alpha = 90°$.

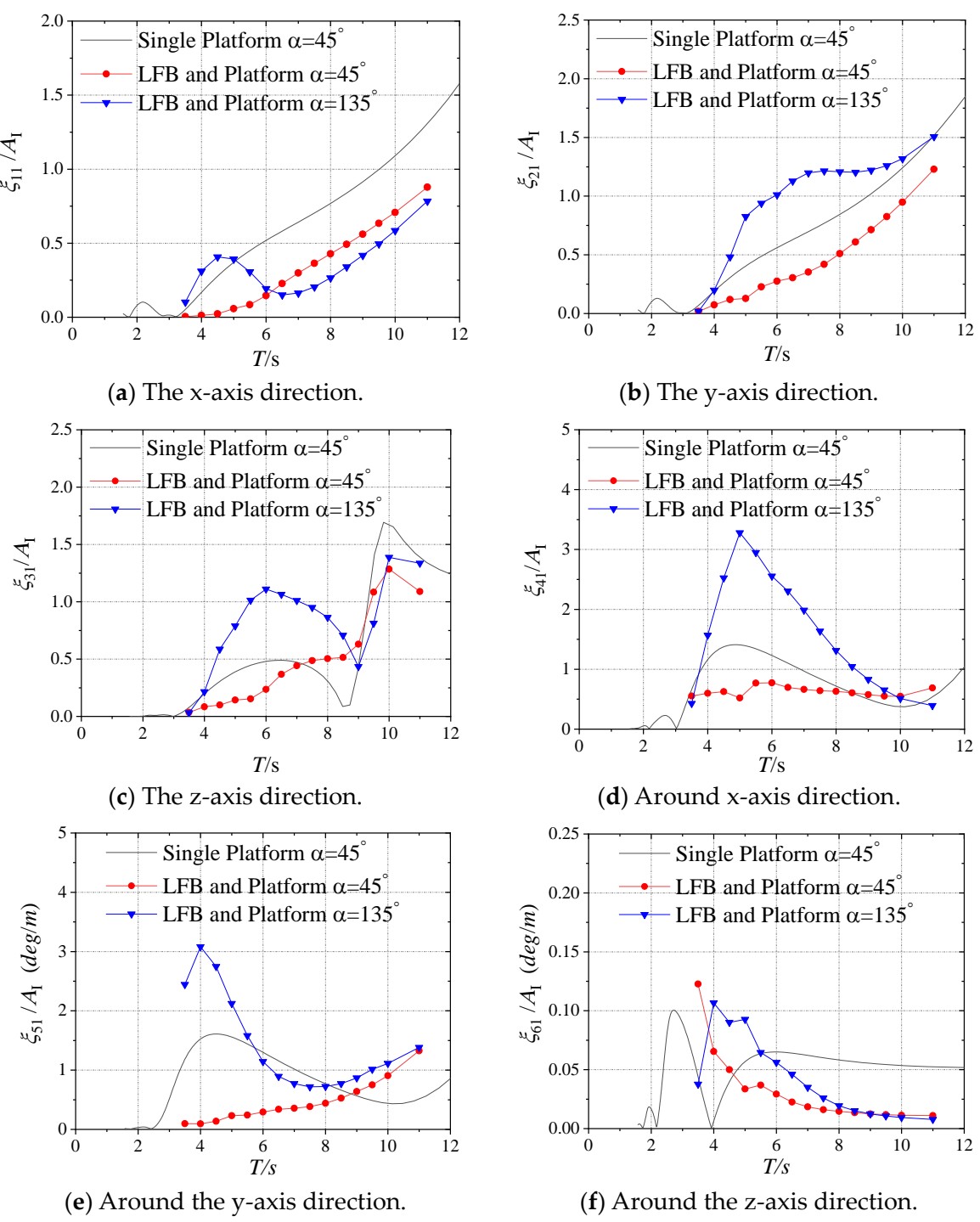

(**a**) The x-axis direction.

(**b**) The y-axis direction.

(**c**) The z-axis direction.

(**d**) Around x-axis direction.

(**e**) Around the y-axis direction.

(**f**) Around the z-axis direction.

**Figure 10.** Motion responses of the small platform at wave angle $\alpha = 45°$ and $\alpha = 135°$.

(1)    The wave-incidence angle $\alpha = 0°$

In Figure 8, the motion responses of the small platform are depicted as the wave period varied at the wave-incidence angle $\alpha = 0°$. The fixed large floating box was positioned in the head wave, while the small platform was located on the back wave side. The results indicate that the motion amplitudes in the $x$-axis direction and $y$-axis rotation were significantly lower than the results of the single platform without the floating box. However, the motion amplitudes around the $y$-axis direction were greater, as the wave periods ranged from 9.0 s to 11.0 s. Additionally, the motion amplitudes in the $z$-axis direction were slightly smaller than the results of the single platform without the floating box when the wave

periods ranged from 3.5 s to 7.5 s and from 9.5 s to 11.0 s. These outcomes indicate the significant shielding effect of the large floating box on the small semi-submersible platform in the *x*-axis direction and around the *y*-axis direction when compared with the results of the independent platform under wave action. However, the shielding effect of the large floating box in the *z*-axis direction was weak.

(2) The wave-incidence angle $\alpha = 90°$

The motion responses of the small platform at the wave incidence angle $\alpha = 90°$ are shown in Figure 9. The wave propagated longitudinally along the y-axis direction of the platform and the floating box. The small platform's motion amplitudes with the fixed floating box in the *y*-axis direction, *z*-axis direction, and around the *y*-axis direction demonstrate a close match with the results of the single platform without the floating box. This implies that the presence of the fixed floating box had minimal effects on the hydrodynamic characteristics of the small platform, specifically in relation to the longitudinal propagation of the waves.

(3) The wave-incidence angle $\alpha = 45°$ and $\alpha = 135°$

Figure 10 illustrates the motion responses of the small platform at the wave-incidence angles of $\alpha = 45°$ and $\alpha = 135°$. The fixed floating box was positioned in the head wave, while the small platform was situated on the back wave side when the wave-incidence angle was $\alpha = 45°$. From the figures, it is evident that the motion amplitudes in the *x*-axis direction, in the *y*-axis direction, around the *x*-axis direction, and in the *y*-axis direction were notably lower than the results obtained from the single platform without the floating box under the same wave-incidence angle $\alpha = 45°$ (Figure 10a,b,d,e). Meanwhile, the motion amplitudes in the *z*-axis direction were slightly reduced compared to the results from the single platform without the floating box, particularly for the wave periods from 3.5 s to 7.0 s and from 10.0 s to 11.0 s in Figure 10c.

The motion amplitudes in the *x*-axis direction, in the *y*-axis direction, around the *x*-axis direction, and in the *y*-axis direction were significantly greater than the results obtained from the single platform without the floating box under the wave-incidence angle $\alpha = 135°$ in Figure 10a,b,d,e. This outcome was due to the fact that the small platform was positioned in a head wave, which caused the fixed floating box to reflect the incident wave, leading to an increase in the wave height on the small-platform side.

It is noteworthy that the presence of the fixed floating box effectively reduced the motion amplitude in the *x*-axis direction, irrespective of the wave-incidence directions, except for when the wave periods ranged from 3.5 s to 5.0 s and the wave-incidence angle was $\alpha = 135°$. These findings suggest that the shielding effect of the fixed floating box on the small platform was significant when the former was in the head wave and the latter was situated on the back wave side, leading to smaller motion amplitudes in the small platform.

### 4.4. Coupled Analysis of Hydrodynamic Responses of Two Floating Bodies

The objective of this section is to investigate the hydrodynamic coupling responses between the small semi-submersible platform and the large floating box, which were constrained by the linear stiffness matrix. In particular, it takes into account the impact of the diffraction and radiation waves generated by the large floating box on the motion responses of the small platform while disregarding the connection between the two floating bodies.

The damping values of the large floating box were obtained based on the maximum value of the radiation damping of the floating box. These parameter settings were not based on an actual model and were only applied in the calculation of examples. The damping changes to the floating box may have exerted a significant impact on the motion responses of the floating box itself and of the semi-submersible platform.

Therefore, the damping matrix of the large floating box $\mathbf{B}_2$ is

$$
\mathbf{B}_2 = \begin{bmatrix}
5.0 \times 10^6 & 0 & 0 & 0 & 0 & 0 \\
0 & 7.0 \times 10^5 & 0 & 0 & 0 & 0 \\
0 & 0 & 5.0 \times 10^6 & 0 & 0 & 0 \\
0 & 0 & 0 & 2.5 \times 10^9 & 0 & 0 \\
0 & 0 & 0 & 0 & 2.5 \times 10^6 & 0 \\
0 & 0 & 0 & 0 & 0 & 3.0 \times 10^9
\end{bmatrix}
\tag{22}
$$

The stiffness matrix of the large floating box $\mathbf{K}_2$ is

$$
\mathbf{K}_2 = \begin{bmatrix}
5.0 \times 10^5 & 0 & 0 & 0 & 0 & 0 \\
0 & 5.0 \times 10^5 & 0 & 0 & 0 & 0 \\
0 & 0 & 0 & 0 & 0 & 0 \\
0 & 0 & 0 & 0 & 0 & 0 \\
0 & 0 & 0 & 0 & 0 & 0 \\
0 & 0 & 0 & 0 & 0 & 1.0 \times 10^7
\end{bmatrix}
\tag{23}
$$

The still-water-restoring-force matrix of the large floating box $\mathbf{C}_2$ is

$$
\mathbf{C}_2 = \begin{bmatrix}
0 & 0 & 0 & 0 & 0 & 0 \\
0 & 0 & 0 & 0 & 0 & 0 \\
0 & 0 & 1.32 \times 10^7 & 0 & 0 & 0 \\
0 & 0 & 0 & 8.82 \times 10^9 & 0 & 0 \\
0 & 0 & 0 & 0 & 1.45 \times 10^8 & 0 \\
0 & 0 & 0 & 0 & 0 & 0
\end{bmatrix}
\tag{24}
$$

4.4.1. Wave-Incidence Angle $\alpha = 0°$

(1) Motion responses of the small semi-submersible platform

Figure 11 presents the motion amplitudes of the semi-submersible platform berthed alongside the large free-floating box, which are subsequently compared to the motion responses of the same single small platform under wave action and to the motion responses of the same small semi-submersible platform docked at the same large fixed floating box under wave action.

The results reveal a few variations between the surge motions of the small platform docked at the large fixed floating box and the outcomes of the small platform docked at the free large floating box in Figure 9a. Nevertheless, a significant difference was observed between the surge motions of the small platform docked at the free large floating box and the motion responses of the single small platform under wave action. This difference indicates that the large floating box had significant shielding effects on the surge motion of the small platform when it berthed at the large floating box with a wave-incidence angle of $\alpha = 0°$.

It can be seen that the heave motion of the small platform docked at the free large floating box was weaker than that of the small platform docked at the fixed large floating box when the wave period was between 5.0 s and 8.0 s. However, the heave motion of the small platform docked at the free large floating box was stronger than that of the small platform berthed at the fixed large floating box and the motion responses of the single small platform under wave action when the wave period was greater than 8.0 s. These findings suggest that when the large floating box was present and the wave period ranged from 5.0 s to 8.0 s, the heave-motion amplitudes of the small platform decreased. However, when the large floating box was present, and the wave period was greater than 8.0 s, the large floating box did not have a shielding effect; instead, it caused greater heaving-motion responses from the small platform.

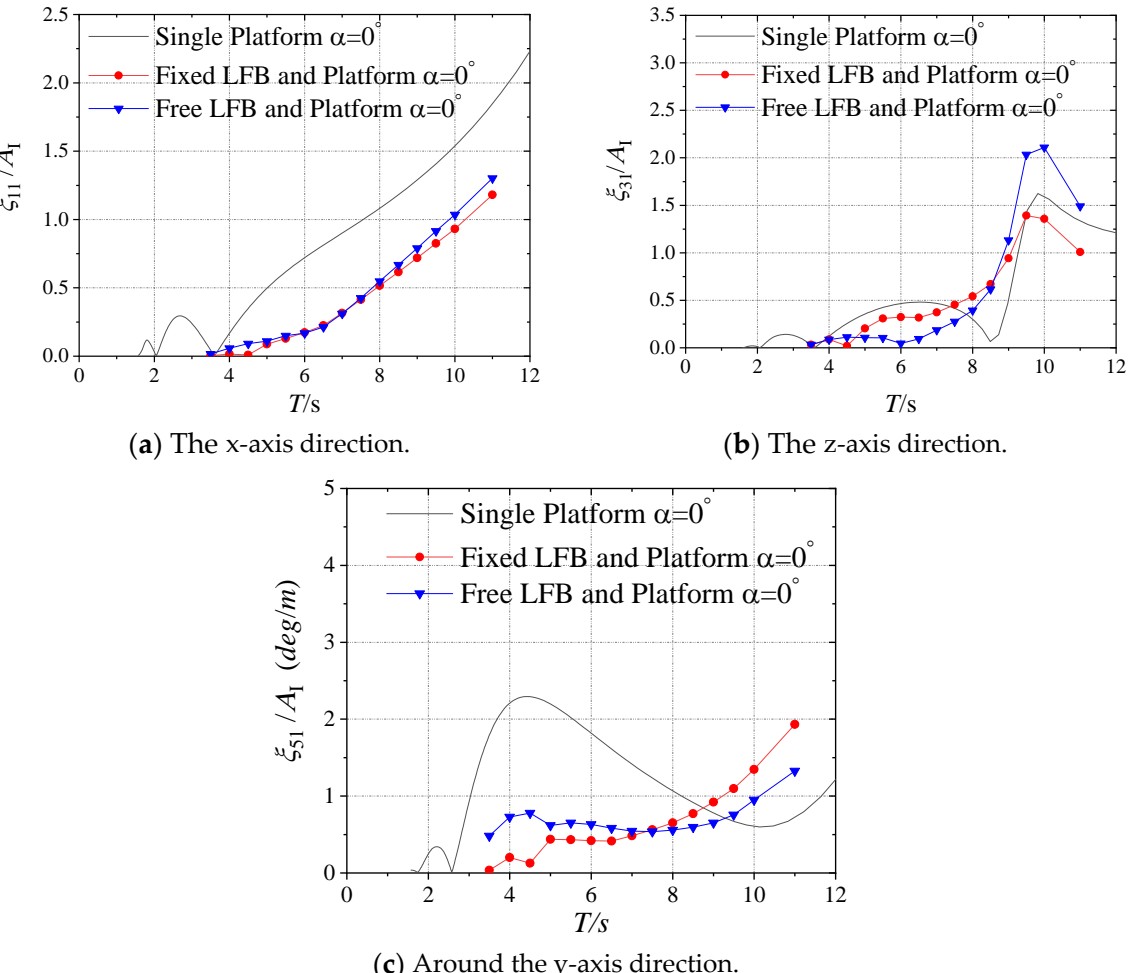

**(a)** The x-axis direction.

**(b)** The z-axis direction.

**(c)** Around the y-axis direction.

**Figure 11.** Motion-response comparison at wave angle $\alpha = 0°$.

The pitch motion of the small platform docked at the free large floating box presented a similar pattern to the heave motion observed in Figure 11c. The pitch motion of the small platform docked at the free large floating box was slightly stronger than that of the small platform docked at the fixed large floating box and substantially weaker than the motion responses of the single small platform under wave action when the wave period ranged from 3.5 s to 7.0 s. However, when the large floating box was present, and the wave period was greater than 9.0 s, the large floating box did not have a shielding effect; instead, it caused greater pitch-motion amplitudes in the small platform.

When the wave period was greater than 7.5 s, the large box had little shielding effect on the vertical motion of the small platform. Instead, the large box amplified the vertical-motion amplitudes of the platform due to an increase in the interactions between the large box and the waves.

(2) Motion responses of the large floating box

Figure 12 displays the motion responses of the large floating box coupled with the semi-submersible platform, which are then compared with the motion responses of the single large floating box under the wave. The results show that the motion responses of the large floating box with the semi-submersible platform agreed well with the outcomes of the single large floating box under the wave. This finding suggests that the small platform had minimal impact on the motion responses of the large floating box when the small semi-submersible platform docked at the large floating box with a wave-incidence angle of $\alpha = 0°$.

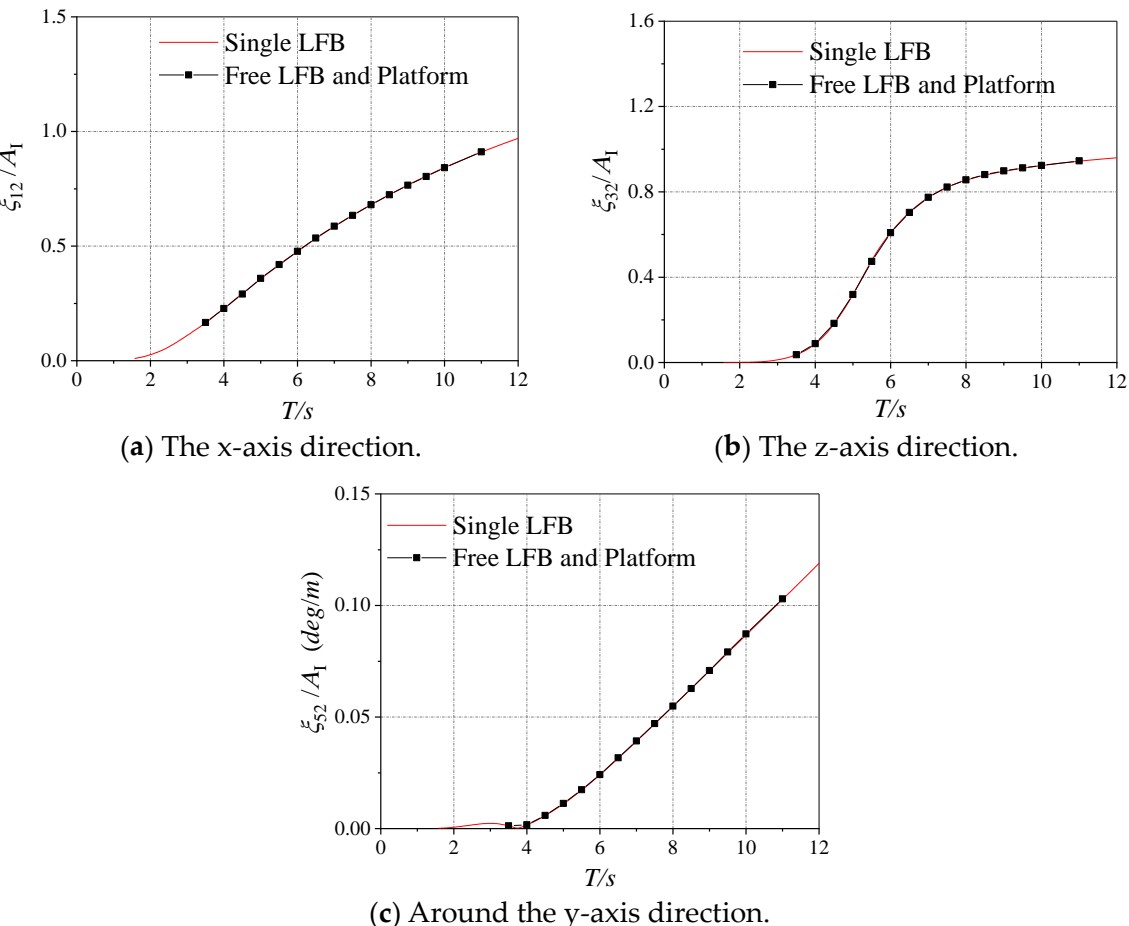

**Figure 12.** Motion response of the LFB at wave angle $\alpha = 0°$.

(3) Relative motion between the small platform and large floating box

The relative-motion amplitudes between the small platform and the large floating box are shown in Figure 13. The $\bar{\bar{\zeta}}_i$ ($i = 1, 2, 3, \ldots, 6$) represents the relative motions in six freedom degrees.

The results demonstrate that when the wave period was below 7.5 s, the relative surge-motion amplitudes between the small platform and the free large floating box exceeded the amplitudes between the small platform and the large fixed floating box. However, when the wave period was greater than 9.5 s, the relative surge-motion amplitudes between the small platform and the free large floating box fell below the amplitudes between the small platform and the large fixed floating box. This indicates that the large fixed floating box provided a better shielding effect than the free large floating box.

Moreover, Figure 13b illustrates that the relative heave-motion amplitudes between the small platform and the free large floating box were higher than the amplitudes between the small platform and the large fixed floating box, primarily as the wave period increased. This means that the large fixed floating box exhibited a better shielding effect than the free large floating box.

When the wave period fell below 7.5 s, the relative pitch-motion amplitudes between the small platform and the free large floating box surpassed the relative motion amplitudes between the small platform and the large fixed floating box in Figure 13c. Conversely, when the wave period exceeded 7.5 s, the comparison reverses = d, indicating that the shielding effect of the free large floating box improved with the increase in the wave period for the pitch motion.

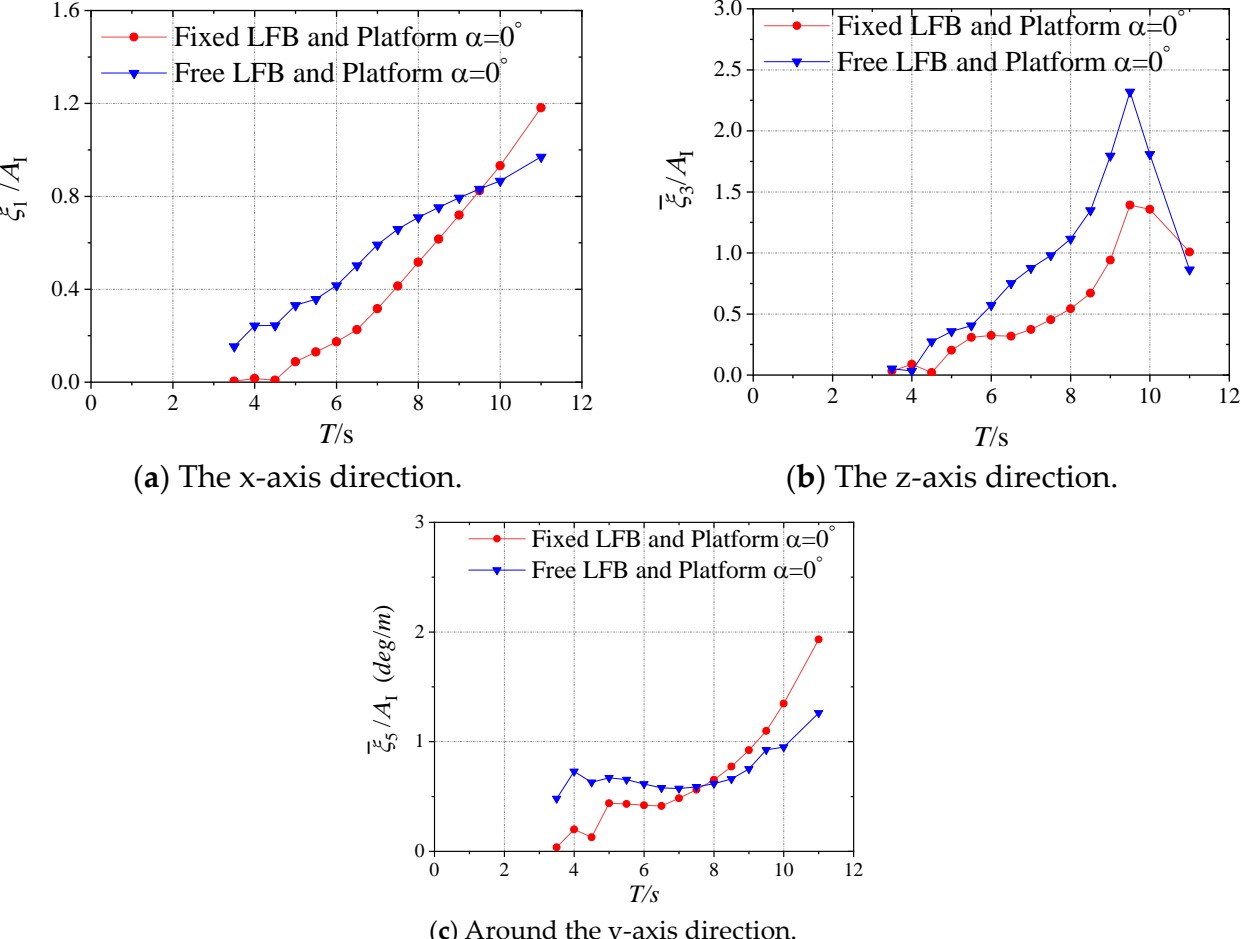

(**a**) The x-axis direction.

(**b**) The z-axis direction.

(**c**) Around the y-axis direction.

**Figure 13.** Relative-motion amplitude between the platform and LFB at wave angle $\alpha = 0°$.

4.4.2. Wave-Incidence Angle $\alpha = 90°$

(1) Motion responses of the small semi-submersible platform

Figure 14 shows the motion amplitudes of the semi-submersible platform docked at the free large floating box and compares them to the motion responses of the same single small platform under wave action and of the same small semi-submersible platform docked at the large fixed floating box under wave action. It shows that the motion responses of the small platform berthed at the large fixed floating box corresponded well with the outcomes of the small platform docked at the free large floating box, as shown in Figure 14.

However, slight variations occurred between the surge motions of the small platform docked at the free large floating box and the motion responses of the single small platform under wave action. This suggests that the large floating box did not have a significant shielding effect on the motion responses of the small platform when the small platform docked at the large floating box with a wave-incidence angle of $\alpha = 90°$.

(2) Motion responses of the large floating box

Figure 15 shows the motion responses of the large floating box with the semi-submersible platform and compares them to the motion responses of the single large floating box under wave action. The results indicate that the motion responses of the large floating box with the semi-submersible platform agreed well with the outcomes of the single large floating box under the wave.

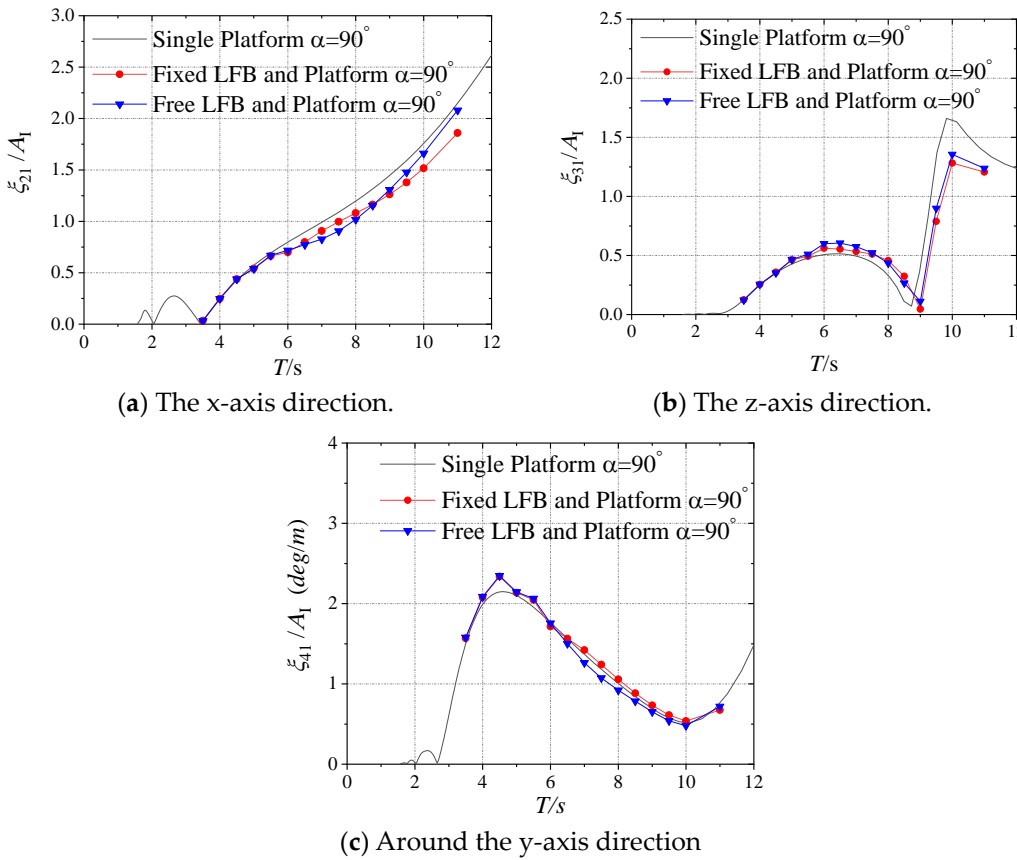

(**a**) The x-axis direction.

(**b**) The z-axis direction.

(**c**) Around the y-axis direction

**Figure 14.** Motion-response comparison at wave angle $\alpha = 90°$.

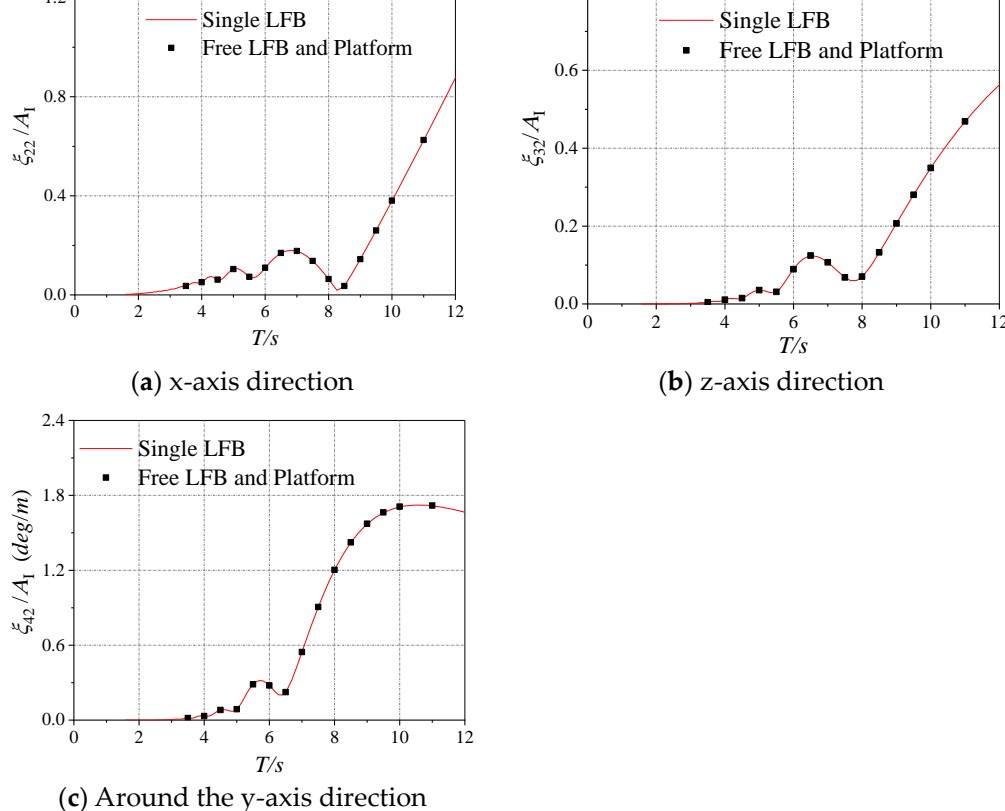

(**a**) x-axis direction

(**b**) z-axis direction

(**c**) Around the y-axis direction

**Figure 15.** Motion response of the LFB at wave angle $\alpha = 90°$.

This finding suggests that the small platform had little impact on the motion responses of the large floating box when the small semi-submersible platform docked at the large floating box with the wave-incidence angle of $\alpha = 90°$.

### 4.5. Analysis of Wave Force of the Semi-Submersible Platform

To analyze the relationship between the wave period and the motion responses of the semi-submersible platform, the distribution characteristics of the wave force when the wave period changed were investigated.

Figure 16 shows the wave forces of the semi-submersible platform during different periods when the semi-submersible platform berthed alongside the fixed floating box and the wave angle was 0, and compares them with the values for the single semi-submersible platform and the semi-submersible platform berthed alongside the free-floating box.

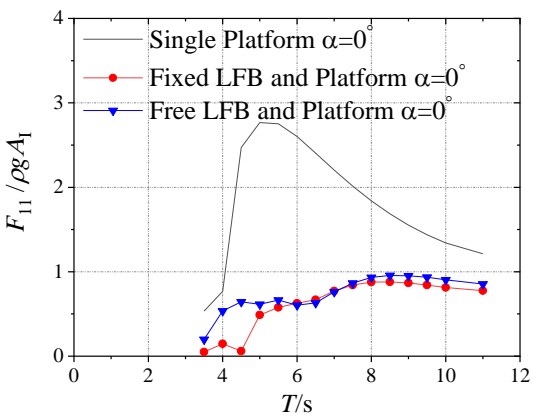
(**a**) Wave force in the x-direction.

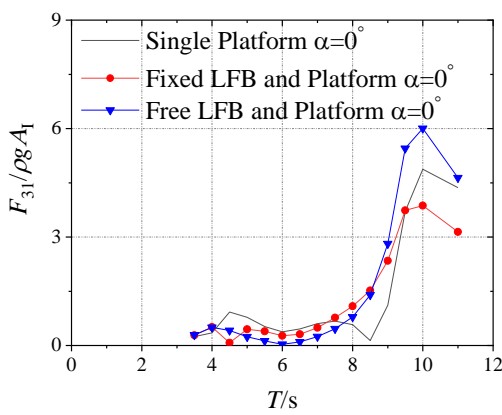
(**b**) Wave force in the z-direction.

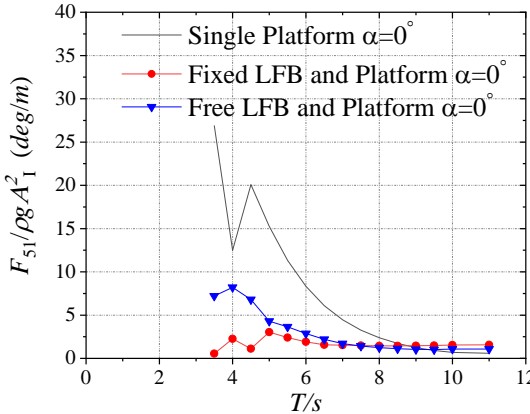
(**c**) Wave moment around the y-axis direction.

**Figure 16.** Wave-forces-and-moments comparison at wave angle $\alpha = 0°$.

It can be seen that as the wave period increased, the trend in the wave force variation was consistent with the motion response. This might explain the characteristics of the motion responses in some specific periods. For example, when the wave period was 4.5 s, the wave force or moment in different directions was very small, and the motion responses of the platform were also very small. When the wave period was greater than 7.5 s, the motion responses of the platform berthed alongside the floating box in the z-direction were greater than those of the single platform because the variation characteristics of their wave forces were same.

The presence of the floating box on the windward side caused significant changes in the wave forces and moments acting on the semi-submersible platform, which, in turn, led to similar changes in the motions of the platform. Other factors, such as the ratio of the

length of the floating box to its width, the connection type, and the spacing between the box and the platform, will be further studied to investigate the essential changes in the motion responses of semi-submersible platforms.

## 5. Conclusions

A direct time-domain high-order boundary-element method using the linear-potential-flow theory was employed to investigate the hydrodynamic-coupling problem between a small semi-submersible platform and a large floating box under wave action. The following conclusions are drawn:

(1) The shielding effect was significant when the large floating box was on the windward side. When the small platform berthed at the fixed floating box or the free-floating box, the surge- and pitch-motion responses of the small platform were smaller than those of the single platform. The heave-motion responses of the small platform were weaker than those of the single platform when the wave period was less than 7.5 s. Above this range, the large box had almost no shielding effect on the heave motion of the small platform, but raised the heave-motion amplitudes instead.

(2) When the large floating box and the small platform were both on the windward side (with a wave-incidence angle $\alpha = 90°$), the large fixed box had almost no shielding effect on the motions of the small platform. The results of the small platform were similar to those of the single platform under wave action.

(3) Compared with the relative-motion amplitudes, when the wave-incidence angle $\alpha = 0°$, the heave-motion amplitudes between the platform and the large fixed box were lower than those between the platform and the large free-floating box. Therefore, additional attention needs to be paid to the relative heave motion when a small platform berths at a large free-floating body. This is especially significant for the jettyless floating transfer system used for connecting LNG transport ships and small transfer platforms via aerial jumper pipes.

Overall, the study highlights the importance of considering the interactions between small semi-submersible platforms and large floating boxes when designing floating transfer systems. It also emphasizes the significance of the wave-incidence angle and the size difference between platforms in determining hydrodynamic responses.

**Author Contributions:** Conceptualization, J.Y. (Jianye Yang) and J.C.; Methodolody, J.Y. (Jianye Yang); Software, J.Y. (Jianye Yang); Formal analysis, J.Y. (Jianye Yang); Supervision, J.Y. (Jun Yan) and Y.Z.; Writing—Original, J.Y. (Jianye Yang); Writing—Review & Editing, Y.Z. and H.J.; Funding acquisition, J.Y. (Jun Yan); Project administration, J.C. All authors have read and agreed to the published version of the manuscript.

**Funding:** This research was supported by the Key Research and Development Program of Ningbo (grant no. 2022Z061 and grant no. 2023Z055), National Natural Science Foundation of China (grant nos. 52001276), Liaoning Province's Xing Liao Talents Program (grant no. XLYC2002108) and Dalian City Supports Innovation and Entrepreneurship Projects for High-level Talents (grant no. 2021RD16).

**Institutional Review Board Statement:** Not applicable.

**Informed Consent Statement:** Not applicable.

**Data Availability Statement:** Not applicable.

**Conflicts of Interest:** The authors declare no conflict of interest.

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
