# Peer review of "Coupled Analysis of Hydrodynamic Responses of a Small Semi-Submersible Platform and a Large Floating Body"

_jmse, doi:10.3390/jmse11071451_

Round 1
Reviewer 1 Report
This paper presents a comprehensive and advanced model and apply it for a practical case. I find the need for several important clarifications and data completion.
(1) The use of the term leeward side in the abstract is confusing. In the conclusion the author use windward (exposed) side, which is the opposite of leeward (protected side). Please check and clarify the abstract.
(2) The paper presents the mathematical formulation of the integral equation for the BEM, however not the numerical formulation. Please add the numerical formulation: the BEM interpolation functions, the singular integration (for the diagonal coefficients), the time domain stepping and all the formulation needed to program the model.
(3) The Green function Rankin’s source with mirror plane at seabed satisfy the seabed boundary condition so it is not included in the mesh. What about the Radiation Condition? I guess the artificial damping at the outer ring (line 232) satisfies it. Please clarify and present the mathematical and numerical formulation of the artificial damping.
(4) In Table 1, correct Breath to Breadth.
(5) The input parameters need to enable repetition of the results by other researchers. Important parameters are missing in Table 1: Breadth and Height of a single Submerged Pontoon, Place of the Columns along the Pontoons. Al the other parameters needed for running the model (in case I missed any). Please also add an Engineering Drawing with the dimensions of the Semi-submersible.
(6) Please, formulate the method to obtain the Viscous Damping Matrix B for equation (16).
(7) Does K represent the mooring of the Semi-submersible to the Barge? Please specify the parameters of the mooring lines (diameter, material, stiffness, length, angle) and the deriving of the linear spring coefficients K.
(8) For the Heave Hydrostatic restoring spring I obtain: The Area of 4 columns of diameter 0.6m is 1.13m2, The Spring coefficient in seawater is 1025 x 9.81 x 1.13 = 11,362 N/m.
Equation (18) specify 44,300 N/m. Maybe the Radius of a column is 0.6m?
No comments.
Reviewer 2 Report
The authors carry out coupled analysis of hydrodynamic responses of a small semi-submersible platform and a large floating body.
The subject may be informative practically. On the other hand, however, as a research paper that could be published in a relevant journal, I cannot figure out what are new in the authors’ work. The conclusions mentioned at the end of the manuscript are, I would say, matters of course. For example, the sentence in the conclusions (1), that is, ‘the existence of the small platform has little effect on the hydrodynamic responses of the large floating box’ is quite obvious without the authors’ work. More examinations from the viewpoint of physics that may be working in the corresponding phenomena are needed in order to be published as a research paper.
There also exist several comments and questions that can be found in the followings, to which, I recommend, the authors pay due considerations and revise the manuscript accordingly.
(1) The examinations of the numerical results on the motion responses mentioned in the manuscript are, I would say, just the enumeration of numerical facts. For example, the examinations mentioned in Lines 449 – 460 just mention the numerical facts. Are the particular numbers such as ‘3.5s to 7.0s’, ‘9.0s’, ‘7.5s’ magic numbers? I would say that they just hold in the particular numerical results conducted by the authors but do not present as to what happens in real structures.
(2) Why do the authors dare to carry out numerical analyses in time domain? I understand, in order to obtain the results presented in the manuscript, frequency-domain analyses are easier.
(3) Are the drift forces acting on the structures accounted for in the analyses? I guess, if drift forces are taken into account, the gap distance between the two structures will slowly vary timewise, which, in turn, should affect the hydrodynamic interactions between the two bodies.
(4) In Lines 170 – 179
It is mentioned that the z, z1 axes pass through the centers of gravity, but, in Figure 1, the axes do not seem to pass the centers of gravity.
(5) The definition of incident angle is not clear. Together with the definition of incident angle, it is better to show graphically the relative locations of the two bodies and the incident wave that were subjected to the corresponding numerical analysis conducted to obtain the results shown in Figure 9 ~ Figure 13.
(6) In Lines 194 – 195
η, SF are missing in the sentence.
(7) In Line 219, 239
I guess ‘time-matching’ should be ‘time-marching’ instead.
(8) In Line 238
It is mentioned that ‘To update the free surface elevation ….’. Does this mean that the free-surface boundary condition was imposed at the instantaneous free surface instead of at the mean free-surface?
(9) In Lines 273 – 274.
I don’t understand what ‘The wave propagates from the box to the cylinder along the long side of the box’ means.
(10) In Lines 303 – 304.
What is the definition of ‘distance between the semi-submersible platform and the large floating body’?
(11) In Lines 405 – 410.
It seems to contradict that in Line 407 it is said ‘incorporates linear elastic constraints’ while in Line 409 it is said ‘while disregarding the connection between two floating bodies’.
(12) In Line 419.
Does ‘the semi-submersible platform docked at a free large floating box’ mean that the gap between the two bodies zero?
(13) In Lines 459 - 460.
The sentence ‘These results may be attributed to alterations in the interaction between the waves and the platform under distinct wave periods’ just mention the numerical fact, which does not present any useful information.
Round 2
Reviewer 1 Report
(1) One important input parameter is still missing: the Height of a single Submerged Pontoon.
No comments.
Reviewer 2 Report
I have no further comments nor questions.
I judge the revised manuscript can now be accepted for publication.